# Decline by design: Assessing decline policies as a decarbonisation strategy under the Paris Agreement

Gregory Trencher[1], Mert Duygan[2,3], Adrian Rinscheid[4], Daniel Rosenbloom[5*], Peter Newell[6]

1 Kyoto University, Graduate School of Global Environmental Studies, Kyoto, Japan, 2 Centre for Development and Environment (CDE), University of Bern, Bern, Switzerland, 3 Department of Environmental Social Sciences, Eawag, Dübendorf, Switzerland, 4 Department of Political Science, University of St.Gallen, St. Gallen, Switzerland, 5 School of Public Policy and Administration, Carleton University, Ottawa, Canada, 6 Department of International Relations, Sussex University, Brighton, United Kingdom

* daniel.rosenbloom@carleton.ca

## Abstract

Limiting climate change to targets set under the Paris Agreement requires urgent action to reduce the production and use of carbon-intensive technologies, fuels, materials and industrial processes. Accordingly, scholars are increasingly studying 'decline policies', which, by design or effect, induce the reduction or discontinuation of carbon-intensive artefacts and activities. However, understanding of the diversity and decarbonisation potential of such policies is hindered by a lack of large-scale, cross-sectoral and cross-national analyses. Here we present a novel dataset of 233 decline policies formulated by the ten highest-emitting Annex I countries to spur decarbonisation. We examine: (1) decline approaches and policies used across sectors; (2) variations in policy design features expected to influence the magnitude of decline and mitigation outcomes; (3) the relationship between decline approaches, policy design features and mitigation impact. We find that decline policies are widely used across the ten countries, including *direct* approaches (targeting incumbent carbon-intensive elements), and *indirect* approaches (promoting substitution with cleaner alternatives). Statistical analysis indicates that policy design is a critical determinant of decarbonisation potential. While evidence that *direct* decline policies might be more effective compared to *indirect* policies is limited, the 'intensity' of decline policies – defined by design features such as strictness, reduction speed and geographic coverage – is significantly associated with higher mitigation impact estimates. Finally, by proposing a typology of archetypical decline policies, our study advances an empirically grounded conceptual framework for understanding decline as a critical strategy for accelerating decarbonisation.

**Data availability statement:** All relevant data are within the paper and its Supporting information files.

**Funding:** The author(s) received no specific funding for this work.

**Competing interests:** The authors have declared that no competing interests exist.

## 1. Introduction

To hold global temperature rise below the levels enshrined in the Paris Agreement [1], there is an urgent need to transform the carbon-intensive technologies and practices contributing to climate change. In addition to policies that promote the development and diffusion of low-carbon innovations, it is increasingly recognised that accelerated decarbonisation also requires 'decline policies', which, by design or effect, catalyse the reduction or discontinuation of carbon-intensive technologies, materials, fuels and industrial processes [2–5]. Reflecting this, among the 'climate clubs' that have emerged in parallel with climate negotiations at United Nations COP meetings, objectives to reduce or phase out coal-fired power, gas and oil extraction, and routine flaring feature prominently [6]. The need for decline policies is also underscored by the International Energy Agency's *Net-Zero Emissions by 2050* scenario, which calls for a rapid reduction of fossil fuels, coal-fired power, fossil-fuel subsidies and vehicles with internal combustion engines (ICE vehicles) by 2030 [7]. Decline policies can take many forms, including phase-outs [8,9], bans [10], laws [11], carbon pricing instruments [12], performance standards [13] and other types of restrictions or governance instruments that induce the reduction or discontinuation of carbon-intensive artefacts, materials and activities.

Alongside these policy developments, scholarly work has seen a surge of theoretical and empirical interest in decline, particularly in the field of sustainability transitions [2–4,14,15]. Seminal papers have examined the decline of the coal industry in the United Kingdom [16], whereas more recent work has covered processes of decline surrounding coal or oil extraction [17,18], coal-fired power [19–21], ICE vehicles [10,11,22,23] and fossil-fuel heating systems [24]. In parallel, political scientists are increasingly foregrounding the importance of supply-side instruments to curb fossil-fuel extraction such as bans, moratoria and subsidy removal [25–27]. This growing focus on deliberate decline responds to a longstanding tendency within the literature to emphasise the 'creation' of innovations [28] at the expense of investigations on the 'destruction' of carbon-intensive technologies and practices [12].

Yet, while substantial progress has been made in elucidating the conditions, policies and processes that contribute to the decline of incumbent industries and technologies, academic work has stopped short of developing a comprehensive and global picture of the extent to which decline policies feature in the contemporary decarbonisation strategies of countries across the world. Indeed, many empirical investigations of decline have drawn from historical cases where climate change concerns were not the central driver of socio-technical change [29–33]. Moreover, the prevailing focus within the literature tends to centre on specific industries, countries or regions [4]. Energy and road transport have received the lion's share of attention, despite the importance of other domains such as waste, buildings, agriculture and land-use not to mention cross-cutting interventions [34]. Expanding the temporal, sectoral, and geographical scope of empirical research is, therefore, critical to deepen understanding of decline.

Beyond questions of scope, additional work is needed to elucidate the distinct role that decline strategies play within policy mixes. Prior work has laid out important conceptual and theoretical foundations for understanding the potentially synergistic

and accelerating effects of policy mixes that simultaneously promote the breakdown of the old and the buildup of the new in the context of pursuing socio-technical transformation and decarbonisation [5,12,28,35,36]. Nevertheless, empirical studies of actual policy mixes that adopt this dual view of innovation and decline remain somewhat limited [12,37].

What is more, the effectiveness of different types of decline policies in reducing GHG emissions – whether assessed ex-ante through modelling or ex-post through empirical verification – has yet to be integrated within such a policy mix perspective on decline and innovation. Although energy system modellers have simulated the mitigation outcomes of various decline promoting policies such as carbon pricing, bans and mandates [38–40], there is still limited understanding of how specific design features – such as policy stringency, scope, timing and reduction ambition – influence decarbonisation effectiveness.

Responding to these interrelated research needs, this study systematically examines the descriptions of climate policies submitted to the UNFCCC by the ten highest-emitting Annex I countries to answer the following questions:

1. What kinds of decline policies can be identified across sectors?

2. How do decline policies differ in terms of their core design features?

3. What is the GHG emissions reduction potential of different policy designs?

To identify quantitative and qualitative trends across countries, we build a first-of-its-kind dataset comprising 233 decline policies used by some of the world's highest emitting economies to pursue decarbonisation under the Paris Agreement.

This study makes three core contributions. First, the original dataset developed herein enables the inaugural assessment of the extent to which various types of decline instruments have been implemented across high-emitting economies. Our analysis covers a wide range of decline policies – including phase-outs, bans, environmental standards/targets and carbon pricing – applied across multiple sectors. Second, we introduce a novel way to conceptualise and classify varying decline policy approaches as well as a 'decline intensity' index to evaluate the design features of policies in our dataset. Extending the policy design literature [41–43], this index provides a structured means of assessing the presence of key policy features expected to influence both the magnitude and pace of decline, as well as decarbonisation outcomes. Third, we provide the first empirical exploration of the GHG reduction potential of different decline approaches and the extent to which policy design may influence mitigation impact. Together, these contributions advance conceptual and empirical understanding of decline as a critical paradigm for pursuing decarbonisation and innovation.

## 2. Conceptual framework

### 2.1. Conceptualising deliberate decline

Within the various literatures engaging with climate policy and decarbonisation, the intentional decline of carbon-intensive systems has emerged as an increasingly salient topic [2–4]. Previous scholarship has characterised the process of deliberate decline as 'the managed erosion of lock-ins that perpetuate the production and consumption of fossil fuels' [2]. Such lock-ins span multiple dimensions, including techno-economic (e.g., sunk investments in carbon-intensive infrastructure), political-institutional (e.g., entrenched incumbent interests or market rules that encourage continued production and use of fossil-fuel technologies), and behavioural (e.g., user norms and practices that perpetuate carbon-intensive lifestyles) [44,45].

The decline of carbon-intensive systems has been extensively examined through the lens of socio-technical transitions [12,46–48]. This tradition conceptualises carbon-intensive systems as large functional systems – for instance, transport, electricity and agri-food – that fulfil critical societal needs such as mobility, energy and food [49]. These systems are stabilised and reproduced by socio-technical regimes, made up of dominant configurations of material elements (technologies, materials, infrastructures) and non-material elements (policies and regulations, business models, user practices) [50,51]. A

paradigmatic example is the socio-technical regime underpinning personal road-based transport. Centred around the ICE vehicle, this regime is supported by material elements such as road networks, refuelling infrastructure and vehicle manufacturing just as much by non-material factors like regulatory frameworks, business models (e.g., loans and leasing), user behaviour (e.g., car-based commuting to work) and wider cultural norms that valorise automobility by associating it with freedom and personal success [52–54]. This socio-technical perspective suggests that relevant targets for decline-seeking actions are not limited to technologies and materials but also extend to infrastructure, industries, institutions and individual practices.

To elucidate the processes contributing to the destabilisation and decline of regimes in large socio-technical systems, much of the transitions scholarship has relied on cases of historical or ongoing shifts [4,30,55,56]. Predominantly focused on cases across Europe and North America, this body of work has explored the decline of coal extraction [18,29], coal-fired and nuclear power [19,21,31,57,58], fossil-fuel heating [59] and legacy transport systems such as tramways and steamships [32,55,60]. Scholars have also examined unsustainable practices such as cloud seeding [61] and fishing [62]. While the observed declines of these historical socio-technical systems largely occurred due to changed market and societal conditions, these studies underscore the pivotal role of public policy in catalysing, accelerating or cementing the discontinuation of unsustainable socio-technical systems or specific elements. This recognition of the importance of public policy aligns with a growing body of scholarship examining the conditions that can accelerate socio-technical transitions, which, as historical analyses show, typically require decades to unfold [35,56,63,64]. This agency-centred perspective, which emphasises decline as a product of *deliberate* policy intervention [9,15], stands in contrast to more processual accounts that tend to view decline as an emergent outcome of broader structural shifts, such as deteriorating economic or technological competitiveness, changed user preferences and societal crises [16,29,31,65]. To be sure, agency has an important role to play in these processual accounts, including through legitimacy struggles surrounding established trajectories and the eventual defection of incumbent players away from these trajectories.

Drawing on this literature, we adopt a broad conceptualisation of deliberate decline, focusing on policies designed to reduce or eliminate one or more core elements of a carbon-intensive system. Consistent with prior research [3,66], our empirical focus includes policies targeting elements such as *technologies and infrastructures* (e.g., ICE vehicles, fossil-fuel boilers, industry equipment), *substances and fuels* (e.g., plastics, ozone-depleting substances), *industry processes* (e.g., landfilling, logging, gas flaring) and *practices* (e.g., flying). Building on earlier conceptualisations of intentional decline [2,67], we define a decline policy as 'a governance intervention intended or expected to induce the partial reduction or complete discontinuation of the production or use of one or more incumbent elements within a carbon-intensive system.'

## 2.2. Direct decline and indirect decline

While the above definition captures policies that *directly* target carbon-intensive system elements for reduction or discontinuation (e.g., legislation to phase out coal-fired power), it is equally important to recognise policies that *indirectly* induce decline by replacing incumbent elements with cleaner alternatives. Examples of such policies include renewable energy portfolio standards and biofuel mandates, which both set percentage targets for substituting fossil-based fuels with a minimum share of renewable energy, or subsidies supporting the removal and replacement of emissions-intensive technologies (e.g., ICE vehicles, incandescent public lighting and fossil-fuel heating systems) with low-emission alternatives. Assuming one-for-one substitution, such policies result in an inverse reduction of the incumbent element by the same proportion as the diffusion that occurs.

To illustrate the relationship between direct and indirect decline policies, we propose a conceptual framework shown in Fig 1. Informed by prior research [2,12,68] emphasising the dialectical yet complementary relationship between decline and innovation dynamics in sustainability transitions, this framework depicts two contrasting but mutually supporting policy approaches through which decline can be *deliberately* pursued. Additionally, this dual view of decline stresses the role of agency and the multiple points from which policymakers can induce socio-technical transitions [9,63,69].

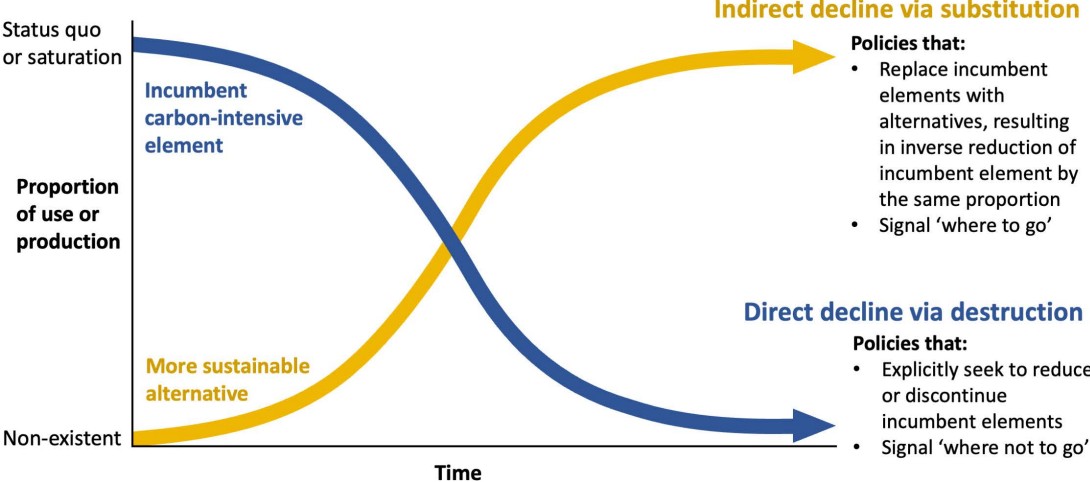

**Fig 1. Direct and indirect decline policies.**

We conceive *direct* decline policies as interventions that reduce the use and/or production of a carbon-intensive element by operating on the downward sloping curve reflecting the relative position of the incumbent element. Direct decline policies thus primarily encompass a destructive function and signal to market actors 'where *not* to go'. Illustrative examples include bans, restrictions or mandates to curtail logging, landfilling and fossil-fuel extraction [70], as well as phase-out schedules for technologies such as coal-fired power [18], incandescent lightbulbs [13,71] or substances like ozone-depleting compounds [72].

Conversely, we conceive *indirect* decline policies as interventions that contribute to decline processes by upscaling alternatives in a fashion which displaces the incumbent carbon-intensive element by an inverse proportion [15,31]. Policies falling into this category include the aforementioned renewable energy portfolio and biofuel mandates as well as ZEV sale targets, which all share the feature of aiming for a specific level of substitution by setting diffusion targets or minimum thresholds. Similarly, policies that mandate or subsidise the replacement of diesel trains, buses or ferries with electric ones, or that promote the substitution of fossil-fuel combustion in industry with electrification or hydrogen, are also expected to result in a proportionate reduction in the use and/or production of the incumbent element. So, if direct decline policies chiefly signal 'where *not* to go', indirect decline policies contribute to the decline process by signalling 'where *to* go,' while at the same time spurring substitution of the incumbent system element.

But not every diffusion policy can be considered an indirect decline policy. This is because many leave room for the uninterrupted continuation of incumbent trajectories, especially in expanding markets where relative decarbonisation gains from a new technology's diffusion can be offset by the absolute growth of the incumbent system. One mechanism that can undermine decline outcomes is *layering*, when a new technology, substance or practice diffuses without displacing or substituting the incumbent element to a comparable degree [73,74]. The case of consumer subsidies for ZEVs provides a compelling example. In some jurisdictions and contexts, ZEV-promoting policies have encouraged the purchase of a second vehicle, thus failing to reduce the size of the existing fleet of traditional ICE vehicles [75,76]. A related mechanism is *rebound*, when per-unit efficiency gains are negated by increased overall use of the incumbent system element. This is particularly relevant for energy-efficiency policies, where rebound effects have been observed to erase gains in reducing fossil-fuel consumption in various domains. In the household sector, this occurs when occupants increase indoor temperatures on their thermostats or when larger dwellings are built [77,78]. Similarly, in the transport sector, many hybrid vehicle owners respond to reduced mile-per-gallon costs by driving longer distances [79].

These concerns about layering and rebound suggest that the potential of certain policy types to reduce the relative share of the incumbent element – whether directly or indirectly – is less certain than other policy types, *ceteris paribus*. Accordingly, as illustrated in Table 1, we limit our empirical scope in this study to policy types where a decline effect can be anticipated with a high degree of certainty. In doing so, we exclude policies such as energy efficiency upgrades and standards, purchase subsidies without replacement requirements, and broad emissions-reduction goals. By narrowing our empirical scope to policies that explicitly perform a *destructive* or *substitutive* function, our conception of two decline approaches distinguishes itself from other work [12,28,80] that take interest in all types of innovation and diffusion policies.

While we do not dismiss the possibility that some of the excluded policy types may, in certain contexts, contribute to the destabilisation or reduction of incumbent carbon-intensive elements, their lack of an explicit destructive or substitutive function places them out of scope from our analysis. With respect to our decision to include all forms of carbon pricing, such as carbon taxes and emissions trading, we follow Kivimaa and Kern's [12] analysis of policy mixes for 'creative destruction', which underscores pollution taxes as critical levers for exerting pressure on emissions-intensive regimes.

A key question arising at this point is which decline approach – direct or indirect – is more effective at spurring decarbonisation and socio-technical transitions. Existing transitions research again provides useful guidance here, suggesting that in most cases successful transitions require both the breakdown of incumbent elements and the emergence of new ones [81,82]. So, while direct decline catalyses the reduction or discontinuation of incumbent arrangements through their destructive function, indirect decline policies contribute to decline through displacement and substitution with the competing technology or practice.

Moreover, indirect decline policies may, at times, prove more socio-politically feasible and easier to justify than more confrontational direct decline policies. Past experiences with direct decline policies targeting established technologies or industries reveal how incumbent power can weaken and delay them [19,83,84]. Thus, to mitigate fears about job losses, asset write-offs and political backlash – challenges repeatedly associated with direct decline policies [18,66,85] – governments

**Table 1. Empirical scope of study and illustrative examples.**

| Encompass an explicitly destructive or substitutive function? | Direct (decline through destruction) | Indirect (decline through substitution) |
|---|---|---|
| **Empirical focus (included)** | | |
| *Yes* | • Bans, moratoria<br>• Phase-out programmes<br>• Restrictions on production/use<br>• Environmental standards<br>• Reduction goals (reducing landfill waste or fertiliser by 30%)<br>• Carbon pricing (taxes, emissions trading) | • Technology replacement program (diesel to electric trains/ships)<br>• Subsidies for fuel switching (gas to electricity) and equipment replacement (oil burners to heat pumps)<br>• Technology upscaling targets (50% renewables by 2030)<br>• Biofuel blending mandates (10% by 2025) |
| **Beyond empirical focus (excluded)** | | |
| *No* | Not applicable | • Energy efficiency upgrades<br>• Fuel efficiency standards<br>• Subsidies to promote ZEVs or renewable energy installation without replacement requirement<br>• General emissions reduction targets<br>• Investments in public transport to encourage modal shift<br>• Building energy efficiency labels |

may choose to embed decline intentions into innovation-focused agendas. For instance, policies within the United States' Inflation Reduction Act introduced under the Biden administration indirectly advanced the decline of fossil fuels, while prioritising job creation, industrial development and economic growth through green industrial policies. This approach underscores the higher political viability of indirect decline in some circumstances, as promises of green technology development and diffusion can temper resistance from incumbent industry, such as lobbying, litigation and political discourse.

### 2.3. An index of 'decline intensity'

A critical determinant of the ability for decline policies – whether direct or indirect – to achieve their intended effect is their design. For instance, policies can differ vastly with respect to their strictness [13,86]. In most cases, mandatory or regulatory instruments offer greater assurance that a particular degree of reduction will occur compared to softer tools like voluntary agreements [87]. Similarly, establishing explicit numerical targets regarding the magnitude of change (e.g., 50% versus 100%) or the speed of change (e.g., immediate, as in the case of some bans, or gradual and multi-decadal, as seen in many phase-out schedules) can play an important role by signalling to the market how much change is required within what timeframe [10,88].

These insights point to the need to consider both quantity and *quality* in studying policy design [41–43]. Take, for instance, the case of a legally enforced schedule that commits a country's power sector to phase-out coal-fired power by 2038, as is the case in Germany. Then compare this to a softer ambition to reduce reliance on fossil-fuel heating by offering financial support for conversion to non-fossil heating or heat pumps through a voluntary programme aimed only at public buildings in a specific region, with no explicit guidance on the market share targeted for reduction. While both policies aim to reduce volumes of the incumbent emissions-intensive technology, the former example of regulatory-backed phase-out at the national scale with a specified schedule can be expected to exert a much larger reduction effect. By the same token, the latter policy can be expected to lead to much larger GHG emissions reductions than policies with weaker decline ambitions or stringency.

Thus, because not all decline policies are created equal, merely counting the number of policies in a particular country or sector may not be a good indicator of the potential decline or decarbonisation effect [86]. Focusing only on quantity while ignoring the qualitative design features of a policy hence risks obscuring variation in policy stringency and undermining the validity of comparative assessments [89]. We deal with this challenge by building on the concept of 'policy intensity' [41] coined by policy design scholars to develop a novel 'decline intensity' index (elaborated in **Methods**). We apply this index to score policies with respect to key design features we expect to influence their ability to achieve large-scale and rapid decline, which also impact the degree of GHG reductions achieved.

## 3. Methods

The principal method employed for this study was the construction, coding and analysis of an original database of decline policies. This database was populated based by examining the contents of Fifth Biennial Reports (the most recent at the time of analysis) submitted by the ten highest-emitting Annex I nations to the UNFCCC in December 2022. As the flagship description of a country's actual or planned mitigation measures, these reports offer a rich and yet-to-be-explored resource for studying decline policies. In what follows, we describe the methodological decisions underpinning our choice of countries, the features of the database, and finally the coding and analytical procedure. The study design is summarised in the four-step procedure illustrated in Fig 2.

### 3.1. Step one: Selection of countries

Annex I parties with emissions reporting obligations under the Kyoto Protocol consist of 44 countries. These collectively represent 44.7% of cumulative GHG emissions globally since 1990 (assigning emissions within the European Union to its member countries to avoid double counting; excluding emissions from land use, land-use change and forestry [LULUCF])

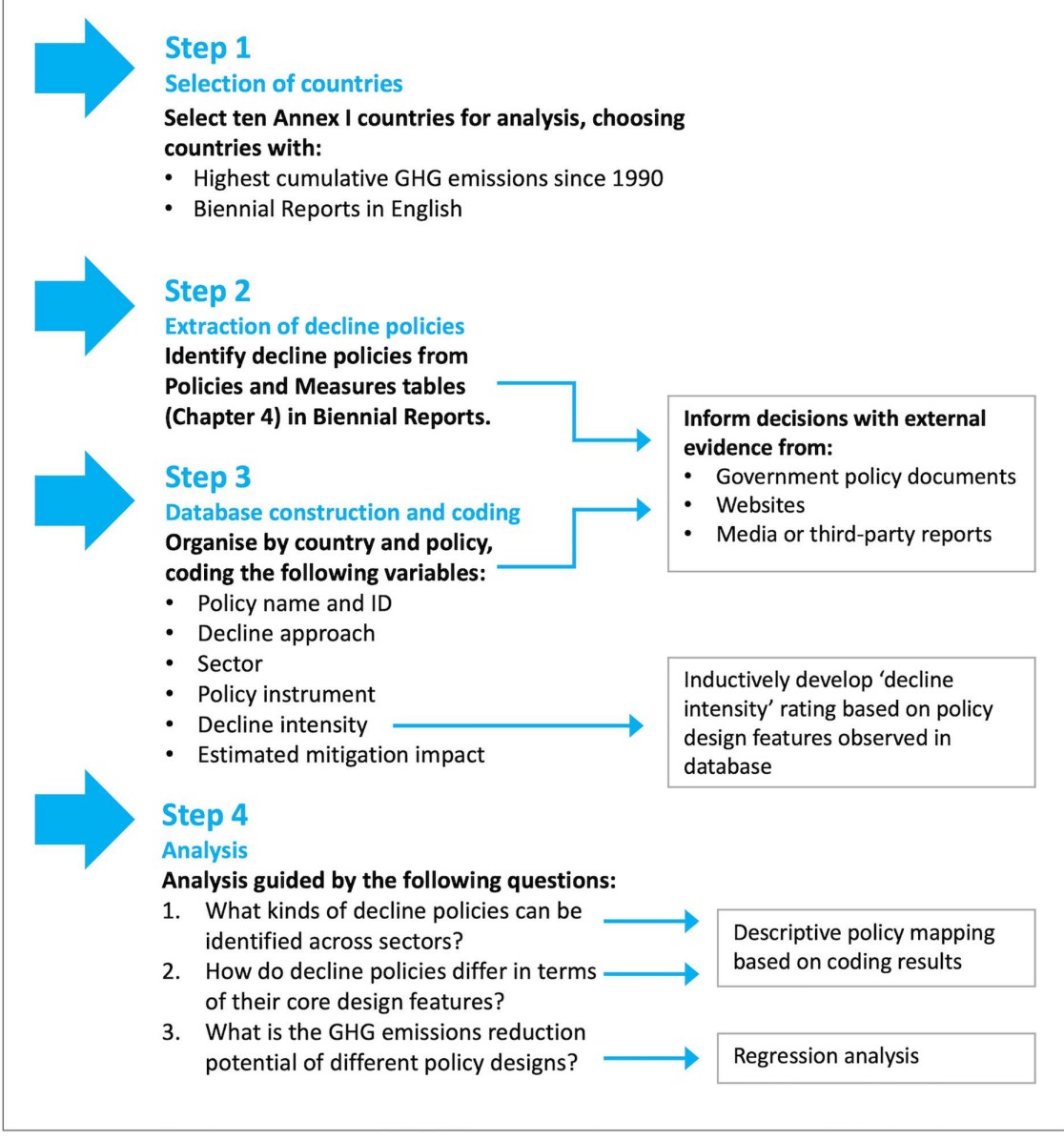

**Fig 2. Research design.**

[90]. Our analysis focuses on the ten Annex I countries with the largest cumulative GHG emissions since 1990, whose Fifth Biennial Reports were available in English. In descending order of emissions, our sample comprises the United States, Japan, Germany, Canada, United Kingdom, Australia, France, Italy, Poland and Kazakhstan. Together, these countries account for 31.8% of global cumulative GHG emissions since 1990 (ibid). The selection of these ten countries allows us to investigate the extent to which decline policies have been implemented in high-emitting and industrialised economies while maintaining a sample size suited to piloting the approach developed herein.

Biennial reports from the European Union (EU), Russia, Spain and Ukraine – countries that would otherwise rank among the top ten global emitters – were excluded from the sample. This decision was made to avoid double counting

the EU with its member countries, to address the unavailability of documents in English from Spain and Russia, and to account for the wartime circumstances in Ukraine, which we expected would hinder the ability to enact decarbonisation policies and collect reliable data on GHG emissions.

### 3.2. Step two: Extract decline polices

Decline policies were identified by manually examining descriptions in each country's 'Policies and Measures' table, along with corresponding explanations in Chapter 4 of the Fifth Biennial Report. We determined the relevance of policies in accord with our conceptualisation in Table 1 and the identification procedure outlined in Table 2. We also provide specific examples of decisions about relevant and irrelevant policies in Tables S1 and S2 in S3 File.

The first question in Table 2 identifies direct decline policies that explicitly state the objective to force or induce the reduction or discontinuation of a carbon-intensive technology, substance/fuel or process. Some policies, such as a regulation or law to phase out coal-fired power, fix this as their primary purpose and are hence included. Additionally, the second question allows us to include policies that induce decline indirectly, primarily through the substitution of incumbent arrangements with new innovations. For example, policies that stipulate an increased market share in an alternative technology encompass an implicit objective to inversely induce the reduction of the incumbent technology, such as in the case of targets to reach a 100% share of ZEVs in new vehicle sales by 2035. It should be noted that we also included broad strategies comprising multiple tenets or measures if one of their focus areas involved decline goals. An example is Canada's 'Green Buildings Strategy'. Although comprising several tenets, such as retrofitting and green construction, we include the strategy due to its explicit ambitions to phase out fossil-fuel heating.

The process of identifying relevant policies was informed by external evidence such as government websites, policy documents and third-party materials. Care was taken to exclude policies that, while described with decline-related terminology (e.g., reduce fossil fuel use) in the Biennial Report, were found upon closer inspection to not correspond with our empirical focus, presented in Tables 1 and 2.

### 3.3. Step three: Database construction and coding

We built a database to compile descriptions of decline policies from each country in accord with variables listed in Table 3. This was organised by country and then structured so that each row corresponds to a specific decline policy and decline target (e.g., coal-fired power, deforestation). The database was populated with information taken directly from the respective Biennial Report regarding the decline target, sector, policy instrument and $CO_2$ reduction estimates in addition to our inductively created coding categories for the decline approach (direct or indirect) and 'decline intensity' (described next). In cases where one policy described several measures addressed at different decline targets, we coded each target separately, assigning each its own sectoral code and decline intensity score. Consequently, the database contains 233 policies that correspond to 251 targets, all of which were included in the analysis. The full database and codebook have been made available in S1 Data, enabling readers to reproduce the analyses conducted for this pilot study.

**Table 2. Criteria used to identify relevant decline policies.**

| Direct decline | 1. Does the policy perform an explicitly destructive function that seeks to reduce/discontinue the incumbent element? | Yes→ Include as a 'direct decline' policy | No→ Proceed to second question |
|---|---|---|---|
| Indirect decline | 2. Does the policy perform an explicitly substitutive function that seeks to replace the incumbent element with a cleaner alternative? | Yes→ Include as an 'indirect decline' policy | No→ Exclude |

Note: Illustrative examples of coding decisions based on these criteria are provided in Tables S1 and S2 in S3 File.

**Table 3. Coding variables used in database.**

| Variable | Example or explanation | Source |
|---|---|---|
| *Decline target (specific and aggregated)* | • Specific (e.g., coal-fired power, heavy-duty ICEs, aviation fuel, gasoline, peat harvesting, deforestation)<br>• Aggregated (fossil-fuel power, fossil-fuel heating, gas use/supply, ozone-depleting substances, transportation fuels) | Inductively created. |
| *Sector* | • Energy supply<br>• Transport<br>• Buildings<br>• Industry<br>• Agriculture<br>• LULUCF<br>• Waste<br>• Cross-cutting | Categories defined by the UNFCCC, self-assigned by countries in their Biennial Report's *Policy and Measures* table. |
| *Approach* | • Direct decline<br>• Indirect decline | Inductively created, explained in Section 2.2. |
| *Policy instrument* | • Regulation<br>• Economic<br>• Fiscal<br>• Information and education<br>• Voluntary<br>• Research<br>• Other | Categories defined by UNFCCC, self-assigned by countries in their Biennial Report's *Policy and Measures* table. |
| *Decline intensity* | Score out of 0, 0.5 or 1 for seven policy design features:<br>1. Strictness<br>2. Numerical targets<br>3. Reduction speed<br>4. Geographic scale<br>5. System coverage<br>6. Reduction magnitude<br>7. Stock turnover | Inductively created (Table 4). |
| *Mitigation impact* | 105 kt $CO_2$-eq in 2030 | Estimates of a policy's impact on annual emissions in a specific year, provided by countries in their Biennial Report's *Policy and Measures* table and supporting tabular data (Table 3). |

Note: The full results of our coding are available S1 Data.

When making coding decisions, we supplemented the descriptions of each policy in that country's Biennial Report with external sources such as official government websites policy documents.

**3.3.1. Evaluation of decline intensity.** Inspired by the concept of 'policy intensity' [41], we developed an original evaluation scheme to compare the varying decline intensity of each policy included in our database (Table 4).

This index was inductively and iteratively designed by observing the contrasting design features of the decline policies in our database. Concretely, we adapted the numerical scoring approach of [41] and [43] to develop seven indicators that evaluate each policy based on a set of design features that we expect to influence the magnitude and speed of reduction and decarbonisation. If an indicator was met, we awarded 1 to that policy, and 0 if unmet or not clear from available evidence. For two indicators (speed and magnitude), we included interim values (0.5) to account for partial fulfilment.

**Table 4. Decline intensity evaluation framework.**

| Indicator | Coding question | Scoring |
|---|---|---|
| (1) Strictness | Is the policy regulatory, legally-binding or backed with a legislation or law? | • 1 if yes<br>• 0 if no or unclear |
| (2) Numerical targets | Is there an explicit target set for either the end year or the magnitude of decline? | • 1 if yes<br>• 0 if no or unclear |
| (3) Reduction speed* | In what timeframe will the reduction occur? | For policies aiming for 51–100% reduction:<br>• 1 if the completion year is ≤ 2035<br>• 0.5 if 2036–2050<br>• 0 if ≥2051<br>For policies with reduction targets between 0–50%:<br>• 0.5 if the completion year is ≤ 2035<br>• 0 if ≥2036; |
| (4) Geographic scale | What is the geographic scale of the policy? | • 1 if national<br>• 0 if sub-national or unclear |
| (5) System coverage | Can the policy be expected to exert a reduction effect on all or most instances of the technology, substance or process and not just a narrow sub-class (e.g., public buildings, a single plant, a single state, low-efficiency plants)? | • 1 if yes<br>• 0 if no or unclear |
| (6) Reduction magnitude | What is the magnitude of reduction aimed for? | • 1 = 100% or ban<br>• 0.5 = 99−50%<br>• 0 = <50% or unspecified |
| **Applied only to decline policies aimed at technologies**** | | |
| (7) Stock turnover | Does the policy induce the reduction of both new and existing installations/instances of the technology? | • 1 if yes<br>• 0 if no or unclear |

Notes: *Bans are interpreted as 100% reduction. ** This indicator was not applied to policies targeting fuel/substances or processes.

As well as using individual scores for each indicator, we also computed 'total intensity scores' by taking the unweighted average score for all relevant indicators (i.e., seven if a policy is targeted at technologies and six if targeted at fuels/substances or processes). Following policy design research [42,43], our underlying assumption is that policies with a higher intensity score would be more likely to achieve large-scale and rapid decline than lower intensity policies, thereby yielding greater decarbonisation outcomes. Although some scholars [41,91] apply weightings to different indicators when calculating policy intensity, we refrained from doing so due to insufficient empirical knowledge about the design features that matter the most for achieving large-scale and rapid decline.

### 3.4. Step four: Analysis

Finally, to build understanding of the decarbonisation potential of different policies and enhance our understanding of associations between their design features and projected mitigation potential, we integrated estimates of yearly mitigation impact, expressed as GHG emissions reductions (kt $CO_2e$) for various years (e.g., 2025, 2030, 2035). Provided in Biennial Reports, these estimates are calculated by governments in each country and were available for 98 (42%) of the 233 policies in the dataset. We excluded three of these policies (ID79, 163 and 207) from the analysis due to implausibly high estimates of mitigation impact that exceed 100% of annual emissions in that sector.

We used regression analysis to model each policy's estimated GHG emission reduction as a function of (a) the total intensity score and (b) the decline approach (direct or indirect). The dependent variable in the regression is the projected mitigation impact of each policy, normalised as a share of total annual emissions in the corresponding sector in 2019. This year was chosen for two reasons. First, it was the most recent year for which emissions data were available in the Fifth Biennial Reports at the time of publication (late 2022). Second, emissions in 2019 were unaffected by the global COVID-19 pandemic, which disrupted emissions patterns from 2020 onwards.

This normalisation was necessary to address substantial variation in the estimates of each policy's mitigation potential and to enable more meaningful comparisons of decarbonisation effectiveness across sectors compared to using unnormalised absolute values. Because the distribution of projected mitigation impacts was highly skewed due to a small number of policies with disproportionately large estimates (ranging between 20–65% of sectoral emissions), we log-transformed the normalised values of avoided sectoral emissions. Due to our interest in the peak mitigation potential of policies irrespective of the timing when this occurs, we used the highest value listed for the policies that provided estimates for multiple years. To account for the hierarchical structure of the data, we applied mixed-effects models with random effects specified at the country level. Finally, to illustrate emission reduction trends in individual sectors, we derived marginal means from the regression results.

Data and Stata code used for the regression are provided in S1 Data and S2 Code.

## 4. Findings

### 4.1. Overview of decline policies in our database

We identified a total of 233 decline policies with 251 corresponding targets across the ten highest-emitting Annex I countries in the UNFCCC. The majority are concentrated in transport, energy supply and waste (Fig 3). Conversely, we found less decline activity in LULUCF and agriculture. Cross-cutting strategies, mostly consisting of carbon pricing instruments such as emissions trading and pollution taxes, make up only 11% of policies. This indicates that policymakers typically use decline policies to tackle well-defined targets in specific sectors through tailored interventions as opposed to a 'shot-gun' style that simultaneously addresses multiple targets and sectors.

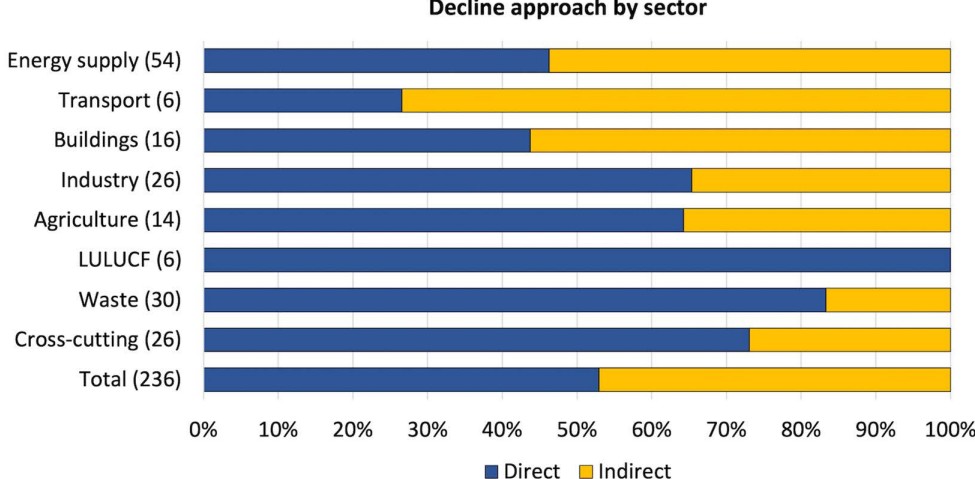

**Fig 3. Distribution of policies across sectors by decline approach.** Bars show the relative distribution of decline policies by sector for 233 policies corresponding to 251 decline targets. The corresponding number of targets for each sector is indicated in brackets. Several policies target more than one sector and thus appear multiple times in the results.

With respect to the distribution of *direct* and *indirect* decline approaches, our analysis reveals a nearly even split comprising 53% and 47% respectively. Sectors with a higher prevalence of direct decline approaches include LULUCF, waste and agriculture. These sectors stand in contrast to those where emissions predominantly stem from fossil-fuel combustion – namely transport, energy supply and buildings – where indirect approaches are more commonly adopted. The transport sector in particular exhibits a strong reliance on indirect approaches, reflecting a tendency to promote substitution with ZEV diffusion targets, fleet replacement programs and renewable fuel blending mandates rather than directly restricting the production or sale of ICE vehicles and petroleum fuels.

Although our analysis did not explicitly aim to identify country-level trends due to the limited size of the Annex I group selected for analysis (n = 10), our findings show that some countries predominantly employ direct decline policies whereas others lean more towards indirect approaches (Fig 4). The three countries with the highest use of direct policies are Germany, Poland and France. Among these, Germany stands out, with more than 75% of its decline policies directly targeting incumbent elements, especially in transport, energy and industry. In contrast, Italy, Japan and the United States exhibit a strong inclination towards indirect approaches.

In terms of the specific emissions-intensive elements targeted by policies, across sectors the most common are fossil-fuel and coal-fired power generation, ICE vehicles (comprising mostly light-duty, but also medium and heavy) and petroleum fuels, fossil-fuel heating, ozone-depleting substances and industrial fossil-fuel combustion, synthetic fertilisers and landfilling of organic waste (see S1 Fig in S3 File). While these technologies, substances and practices are addressed by the majority of the ten countries, our analysis also identified less common but emerging decline targets. The supply of fossil gas in heating and cooking is one example, with four countries (Australia, Canada, UK and France) describing various decline policies at the national or sub-national level. Australia's Capital Territory has taken a particularly active stance, leveraging regulatory instruments to prevent the installation of new gas grids in buildings and urban developments, financial incentives to switch from gas to electric appliances and institutional commitments to using gas-free buildings in the government and private sector.

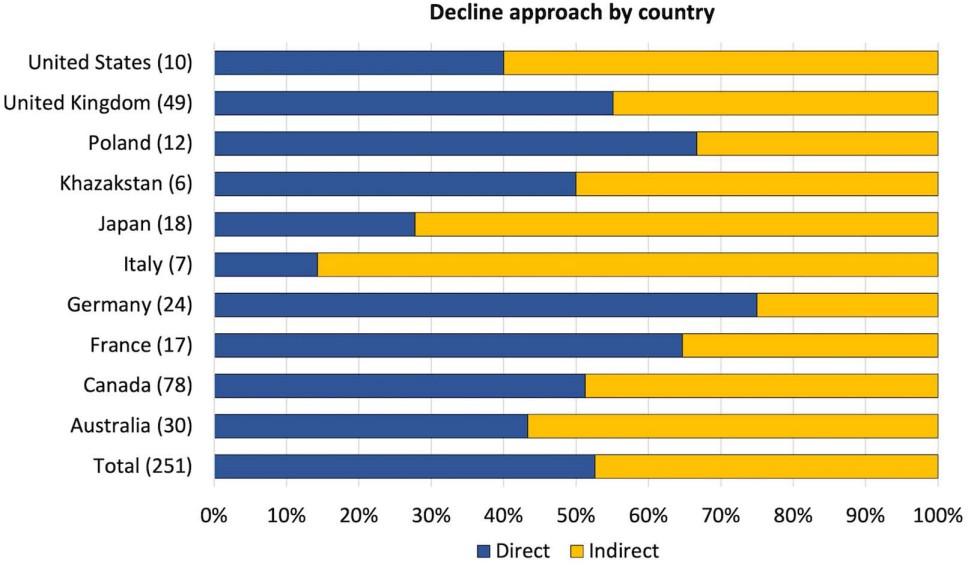

**Fig 4. Distribution of policies across countries by decline approach.** Bars show the relative distribution of decline policies by country for 233 policies corresponding to 251 decline targets. The corresponding number of policies for each country is indicated in brackets.

Tables 5 and 6 provide illustrative examples of *direct* and *indirect* decline policies from our dataset. The examples show that policymakers adopt diverging approaches and instruments to accomplish their aims, including phase-down/out schedules, regulations, prescriptive targets and bans. Consider, for example, how Australia's biennial report lists an indirect decline policy (*Electric Vehicle Strategy*) from the state of South Australia that seeks to achieve 50% sales of ZEVs by 2030 – which inversely signifies a 50% reduction in ICE vehicle sales – while the UK's report describes a regulatory policy (*Regulation to phase out all new non-zero emission road vehicles*) directly banning the sale of ICE vehicles by 2035.

Our analysis reveals distinct differences between the types of policy instruments used by direct and indirect decline approaches. As shown in Fig 5, we find that direct decline policies are more likely to use regulatory instruments than indirect decline policies. Regulatory instruments feature in more than two thirds (69%) of direct decline policies compared to only 36% of indirect decline policies. Indirect decline policies, on the other hand, tend to involve softer strategies such as economic, fiscal and information/education instruments. Many of these instruments involve subsidies or fiscal incentives to encourage the replacement of incumbent technologies like industry equipment and fuels with cleaner alternatives. We also find that the decline policies frequently involve a mix of instruments, with roughly one third (32%) of the policies in our dataset using two or more instruments.

**Table 5. Direct decline: Illustrative policies by sector.**

| Sector | Decline target | Instrument examples | Policy examples (name and ID)* |
|---|---|---|---|
| *Energy supply* | One or more fossil fuel power sources. | • Targets, timelines and environmental standards to reduce or phase out coal power by a specific year. | • Germany's 'Act on the Phase-Out of Coal Plants' (127)<br>• Kazakhstan's 'Reducing the Share of Coal in Power Generation' (246) |
| *Transport* | • ICE vehicles.<br>• Production or use of fossil fuels in road transport. | • Bans on the sale of new ICE vehicles.<br>• Vehicle registration tax based on CO2 emissions.<br>• ICE vehicle-free zones. | • Germany's 'Stronger Weighting of the CO2 Component of the Vehicle Tax from 2021 Onwards' (140)<br>• United Kingdom's 'Regulation to Phase Out all New Non-Zero Emission Road Vehicles' (176)<br>• France's 'Low-Emission Zones (ZFE)' (216) |
| *Buildings* | On-site combustion of fossil-fuels in building equipment. | • Bans on installations of new fossil-fuel heating systems.<br>• Building codes setting rising limits on on-site fossil-fuel use. | • Germany's 'Building Energy Law (GEG)' (134)<br>• United State's 'Federal Building Rulemakings (203)<br>• Canada's (Québec) 'Chauffez Vert Program' (223) |
| *Industry* | • Production/use of ozone-depleting substances.<br>• Fossil-fuel combustion. | • Mandatory targets to reduce ozone-depleting substance production.<br>• Carbon pricing instruments like carbon taxes or emissions trading. | • Australia's 'HFC Management – Regulations' (2)<br>• Poland's 'Proposed Adoption of More Stringent Requirements for the Limitation of the use of Fluorinated Greenhouse Gases' (241)<br>• Kazakhstan's 'Carbon Tax on Sectors not Covered by the Emissions Trading System' (248) |
| *Agriculture* | Fertiliser use. | • Voluntary or mandatory targets to reduce fertiliser use. | • Canada's 'Fertilizer (30% Reduction From 2020 Levels)' (107)<br>• Germany's 'Implementation of German Fertilisation Ordinance 2020' (144) |
| *LULUCF* | Deforestation or land use. | • Bans on deforestation<br>• Aspirational deadlines to end peat harvesting. | • Australia's (Victoria) 'Land use, Land-use Change and Forestry Sector Emissions Reduction Pledge' (28)<br>• Germany's 'Reduction of Peat as Substrate in Gardening' (147)<br>• United Kingdom's 'Forestry Act Felling Licence Regulations and Environmental Impact (Forestry) Regulations' (164) |
| *Waste* | Landfilling of organic or food waste. | • Ban landfilling of organic waste.<br>• Mandates to reduce food waste practices. | • Italy's 'Increase Separate Collection of Urban Waste' (232)<br>• Poland's 'Reduction of Food Losses (Act on the Prevention of Food Waste)' (243) |

*ID numbers refer to specific policies listed in the codebook (see S1 Data).

**Table 6. Indirect decline: Illustrative policies by sector.**

| Sector | Decline target | Instrument examples | Policy examples (name and ID)* |
|---|---|---|---|
| *Energy supply* | One or more fossil fuel power sources. | • Aspirational or mandatory timelines or environmental standards to increase the share of renewable electricity.<br>• Renewable portfolio standards. | • Australia's (Northen Territory) 'Renewable energy target' (18)<br>• Japan's 'Renewables Obligation' (158)<br>• Kazakhstan's 'Increasing the Share of RES in Power Generation' (250). |
| *Transport* | ICE vehicles | • Aspirational or mandatory timelines or environmental standards to increase the share of ZEVs.<br>• Subsidies or vehicle replacement programmes in fleets.<br>• Mandatary targets for biofuel blending in petroleum fuels. | • Canada's 'Light-Duty Vehicles GHG and ZEV Sales Requirements' (74)<br>• Japan's 'Introduction of EV Waste Collection Vehicles' (114)<br>• United Kingdom's (Scotland) 'Vessel Replacement Programme' (189)<br>• Poland's 'Development of Electromobility (Bus Replacement)' (239)<br>• United Kingdom's 'Renewable Transport Fuel Obligation (RTFO)' (155) |
| *Buildings* | On-site combustion of fossil-fuels in building equipment. | • Subsidies for installations to replace fossil-fuel heating equipment. | • United Kingdom's 'Boiler Upgrade Scheme' (173)<br>• Poland's 'Warm Flat Priority Programme' (237) |
| *Industry* | Fossil-fuel combustion | • Subsidies for replacing existing equipment or fuel switching. | • Canada's 'Québec Bioenergy Program' (43)<br>• Japan's 'Promotion of Fuel Conversion' (113)<br>• Italy's 'Incentives to Biomethane and Other Advanced Biofuels' (229) |
| *Agriculture* | Fertiliser use or conventional agriculture. | • Subsidies or capacity building to support conversion to organic agriculture or to natural fertilisers. | • Germany's 'Increase of Organic Farming to 20% of Total Agricultural |
| | | | Utilised Land' (145) |
| | | | • France's 'Agricultural Component of the Circular Economy Roadmap' (210) |
| *LULUCF* | None identified. | None identified | None identified |
| *Waste* | Reduce landfilling waste. | • Subsidies, targets or capacity building to divert landfilling by encouraging or mandating recycling. | • Australia's 'Food Waste for Healthy Soils Fund' (9) |

*ID numbers refer to specific policies listed in the codebook (see S1 Data).

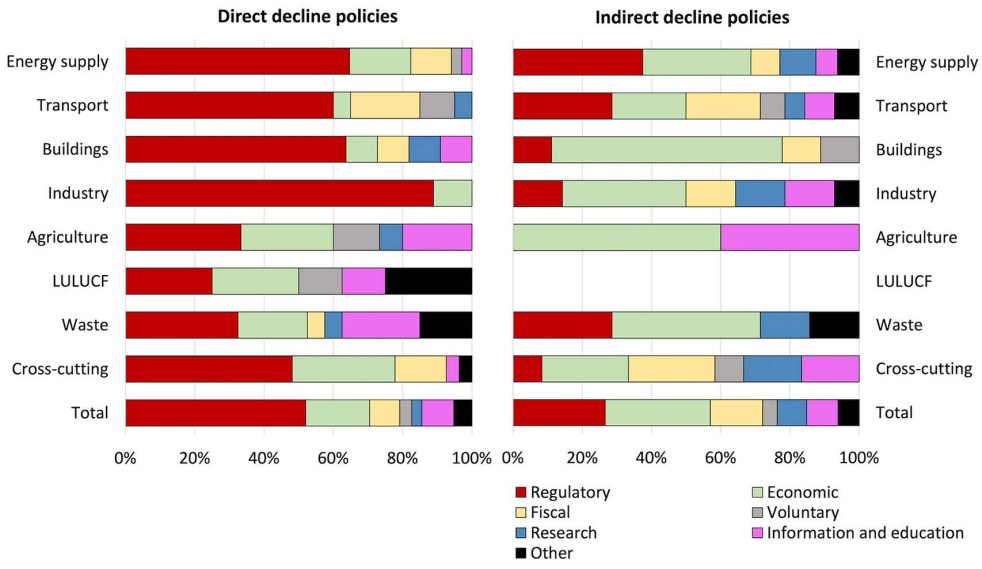

**Fig 5. Relative share of instruments used in direct decline and indirect decline.** Figure shows the relative distribution of all instruments (n = 357) used by all policies (n = 233) following categories indicated in Biennial Reports. The former amount is higher than the latter because roughly one third of policies use two or more instruments.

## 4.2. Variations in decline intensity

We now move to consider the rich variation of design features across policies, assessing their 'decline intensity' in accord with the indicators outlined in our Methods (**Section 3**).

Observing raw means, we find that direct decline policies tend to exhibit a higher decline intensity compared to indirect approaches (Fig 6). This result reflects a stronger tendency for direct approaches to pursue decline through mandatory instruments such as regulation, legislation or legal frameworks (indicator 1). Direct approaches also tend to set explicit numerical targets (indicator 2) and faster reduction schedules (indicator 3). Moreover, they tend to cover larger shares of the incumbent element in the targeted system (indicator 5) and are considerably more likely to aim for deep reduction or complete discontinuation, rather than partial reduction (indicator 6).

Findings also reveal significant variation in the decline intensity of policies across targets (Fig 7). Policies with the highest total decline intensity are typically directed at coal-fired power, deforestation, the production and use of ozone-depleting substances, and the landfilling of organic waste. In contrast, fertiliser and land conversion in agriculture and fossil-fuel based heating in buildings generally exhibit weaker decline intensity scores. This is largely attributable to weak or missing policy design features concerning stringency, explicit numerical targets and articulated ambitions to achieve a complete phase-out.

Across all decline targets, some indicators exhibit particularly high variation in intensity scores – notably strictness (indicator 1), system coverage (indicator 5) and reduction magnitude (indicator 6) – while others are associated with low scores throughout. Some targets achieve high scores on most or all dimensions. In the case of policies targeting coal-fired power, this reflects more frequent use of mandatory frameworks, system-wide coverage, and ambitions to achieve

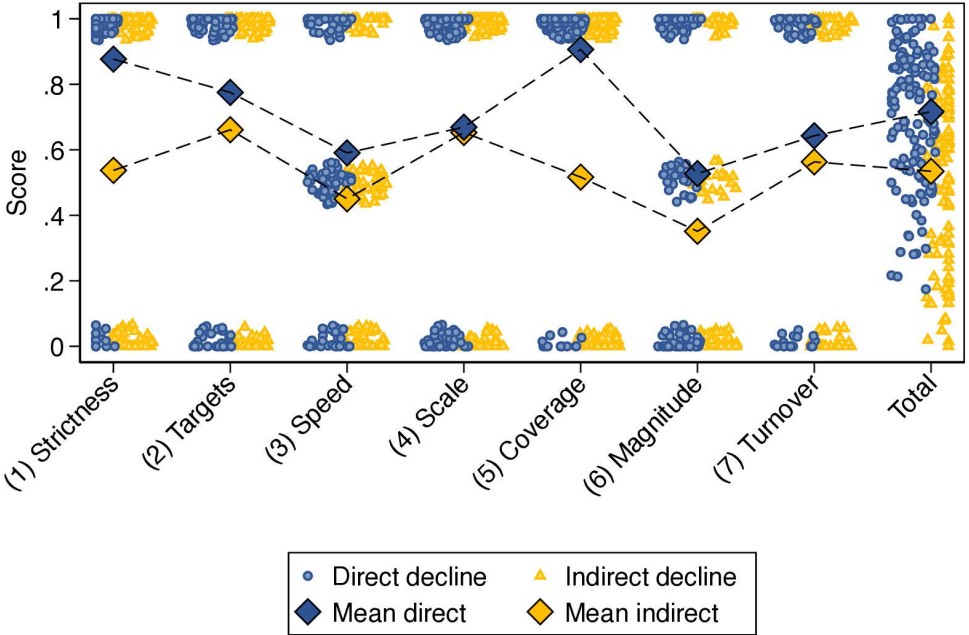

**Fig 6. Decline intensity of direct and indirect approaches.** Jitter plot showing the distribution for seven indicators used to evaluate decline intensity of policies. Circles and triangles represent individual policies, scored according to the criteria elaborated in **Methods** Table 4. Total decline intensity (last column) represents the average unweighted score of all indicators; a policy fully satisfying all indicators would receive a total score of 1. Diamonds represent arithmetic means for all indicators, distinguishing between direct and indirect decline. Indicator 7 (stock turnover) is only applied to decline policies targeting technologies. Based on 233 policies encompassing 251 decline targets. Each target received its own decline intensity score and appears in the figure.

| | (1) Strictness | (2) Numerical targets | (3) Reduction speed | (4) Geographic scale | (5) System coverage | (6) Reduction magnitude | (7) Stock turnover* | Total decline intensity | No. of policies |
|---|---|---|---|---|---|---|---|---|---|
| **Energy supply** | | | | | | | | | |
| Fossil-fuel power | 0.70 | 0.74 | 0.59 | 0.44 | 0.81 | 0.41 | 0.74 | 0.64 | 27 |
| Coal-fired power | 1.00 | 1.00 | 0.86 | 0.43 | 1.00 | 0.79 | 1.00 | 0.87 | 14 |
| Gas use/supply | 0.83 | 0.50 | 0.25 | 0.67 | 0.83 | 0.17 | 0.33 | 0.56 | 6 |
| **Transport** | | | | | | | | | |
| ICE vehicles | 0.61 | 0.83 | 0.65 | 0.65 | 0.41 | 0.63 | 0.26 | 0.59 | 46 |
| Petroleum fuels (road/air) | 0.80 | 0.80 | 0.40 | 0.73 | 0.93 | 0.03 | - | 0.62 | 15 |
| ICE trains/ships | 0.40 | 0.80 | 0.40 | 1.00 | 0.80 | 0.70 | 0.60 | 0.67 | 5 |
| **Buildings** | | | | | | | | | |
| Fossil-fuel heating | 0.57 | 0.48 | 0.36 | 0.67 | 0.48 | 0.45 | 0.43 | 0.50 | 21 |
| Fossil-fuel power use | 0.33 | 1.00 | 1.00 | 1.00 | 0.00 | 1.00 | 0.33 | 0.70 | 3 |
| **Industry** | | | | | | | | | |
| Ozone-depleting substances | 0.88 | 0.69 | 0.38 | 1.00 | 0.81 | 0.38 | - | 0.69 | 16 |
| Fossil-fuel combustion | 0.84 | 0.52 | 0.30 | 0.52 | 0.88 | 0.10 | 0.72 | 0.57 | 25 |
| Venting and flaring | 1.00 | 0.60 | 0.40 | 0.20 | 0.80 | 0.20 | - | 0.60 | 5 |
| **Agriculture** | | | | | | | | | |
| Fertiliser | 0.44 | 0.33 | 0.17 | 1.00 | 0.78 | 0.00 | - | 0.45 | 9 |
| Land conversion | 0.33 | 0.67 | 0.50 | 0.67 | 0.33 | 0.33 | - | 0.47 | 3 |
| **LULUCF** | | | | | | | | | |
| Deforestation | 1.00 | 0.67 | 1.00 | 0.67 | 0.67 | 0.67 | - | 0.86 | 3 |
| Peat harvesting | 0.00 | 1.00 | 0.50 | 1.00 | 0.50 | 1.00 | - | 0.73 | 2 |
| **Waste** | | | | | | | | | |
| Landfilling of organic waste | 0.81 | 0.90 | 0.62 | 0.52 | 0.90 | 0.48 | - | 0.70 | 21 |
| Food waste | 0.50 | 0.75 | 0.56 | 0.75 | 0.88 | 0.50 | - | 0.66 | 8 |
| Plastic | 1.00 | 0.75 | 0.50 | 1.00 | 0.50 | 0.50 | - | 0.71 | 4 |

**Fig 7. Evaluation of decline intensity for most common decline targets.** Results for a subset of the most common targets (n = 233) out of a total of 251 in the dataset. For explanation of indicators, see Methods (Section 3). When viewing score trends across different targets and indicators, readers should note the number of corresponding policies (several policies encompass multiple decline targets) shown in the far-right column. *Stock turnover (indicator 7) is only applied to policies targeting technologies.

complete abolishment. By contrast, policies addressing venting and flaring in oil and gas production or fossil-fuel heating in buildings tend to score relatively low on most dimensions. Petroleum transport fuels, fossil-fuel combustion in industry, fertiliser use in agriculture along with venting and flaring are also notable in that most decline policies aim for partial rather than complete reduction or fail to set explicit reduction timelines. For instance, in the agricultural sector, we identified nine policies across six countries aiming to curb sectoral emissions by reducing fertiliser volumes. Yet only half of these policies are supported by regulatory frameworks, and just three fix numerical targets to specify the pace and magnitude of the reduction.

The above variations in decline intensity are likely due to manifold and context specific factors. Existing research on decline policies emphasises factors such as resistance by industry or affected stakeholders [92,93], the cost and difficulty of eliminating incumbent elements in some sectors [9] and the availability of viable alternatives [94]. This once again foregrounds the critical role of indirect decline policies, which facilitate decline through substitution, by supporting the diffusion of alternatives that replace incumbent carbon-intensive elements. In the case of efforts to reduce landfilling waste, policies promoting alternative means of treating or disposing of waste – such as recycling, composting or incineration facilities – are just as pivotal as policies that force the diversion of waste streams away from landfills.

## 4.3. The decarbonisation impact of decline policies

The preceding analysis highlights substantial variation in the design features of different decline policies. This prompts the question of whether, and to what extent, these design features are associated with differences in decarbonisation effectiveness. To explore this relationship, we draw on estimates of GHG emission reductions (i.e., mitigation impact) anticipated for the implementation of each policy, as reported in each country's Biennial Report. These estimates enable us to examine the mitigation impact of 95 out of the 233 policies in our dataset. We employ multiple regression analysis following the procedure outlined in **Methods 3.4** to sequentially model the relationship between projected mitigation impact (normalised as a share of sectoral emissions) and two variables of interest: (1) the decline approach (direct versus indirect), (2) and decline intensity. Full results of the regression along with correlations of variables are provided in SI Tables S3 and S4 in S3 File.

Our first partial model (n=95), using *decline approach* as the focal predictor variable, reveals a statistically significant relationship (p<0.05) with mitigation impact. This result indicates that direct decline policies tend to be associated with higher estimates of GHG emission reductions. The second partial model, employing *decline intensity* as the primary predictor, shows a robust association (p<0.01) with GHG emission reductions, indicating that higher-intensity policies tend to be linked with larger mitigation impact estimates. In the full model, which includes both independent variables along with sectoral dummies, only decline intensity remains significantly associated with mitigation impact (p<0.01). This is not surprising, given the moderate correlation between approach and intensity (r=0.41; see Table S4 in S3 File) and the relatively small sample (n=95). Substantively, this suggests that the strength and ambition of a policy's design may be more decisive for reducing emissions than the choice of targeting a carbon-intensive element directly or indirectly. Overall, the results provide tentative evidence for our assumption that decline intensity, reflecting the various policy design features discussed in Table 4, matters strongly for decarbonisation outcomes.

As depicted in Fig 8, we next illustrate mitigation potential at the sectoral level by modelling marginal means for both the decline approach (indirect versus direct) and decline intensity (low versus high). To enhance interpretability, we exponentiated the model predictions, allowing the marginal means to be presented on the original scale – that is, as directly interpretable percentage reductions in sectoral emissions. These marginal means reflect the model-predicted average emission reduction associated with a single policy in each sector.

With respect to the decline approach, results (Fig 8, top) indicate that direct decline policies are associated with higher mitigation impact estimates in most sectors, with the exception of energy & buildings and transport. The higher mitigation potential of direct policies is particularly apparent in the industry and waste sectors. Here the average reduction potential for direct decline nears 8% of sectoral emissions compared to 3.3% and 1.5% for indirect approaches, respectively. However, as indicated by the relatively wide confidence intervals, this result is accompanied by a notable degree of statistical uncertainty.

Turning to the relationship between decline intensity and mitigation impact (Fig 8, bottom), we find that in all sectors except waste, high-intensity policies (reflecting a total intensity score >0.5) are associated with greater GHG reduction estimates than low-intensity policies (total intensity score ≤0.5). When considering results for all sectors combined (displayed as grey bars), high-intensity policies are projected to reduce sectoral emissions by an average of 3.8% – considerably higher than the 1.1% reduction estimated for low-intensity policies. The difference between low-intensity and high-intensity approaches is also particularly pronounced in the energy & building sector, where a decline policy with low intensity is predicted to achieve a 0.1% reduction of sectoral emissions, while a high-intensity policy is associated with a reduction of 1.4%.

Taken together, the results of the regression analysis (Tables S3 and S4 in S3 File) and the examination of sectoral marginal means (Fig 8) indicate that policy characteristics are a critical factor shaping the potential depth of decarbonisation achieved by decline interventions. While evidence regarding the relative influence of direct versus indirect decline policies remains inconclusive, the analysis reveals a robust association between the 'intensity' of decline policies and higher emission reduction estimates. These findings therefore suggest that not all decline policies are equally effective. Rather, qualitative variations resulting from design features such as strictness, reduction magnitude, speed and geographic

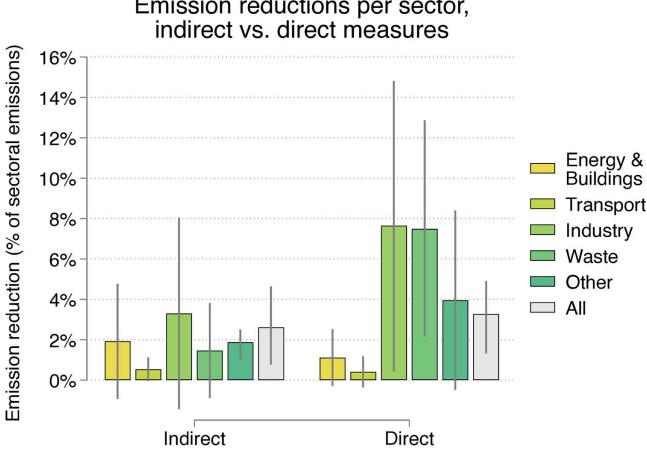

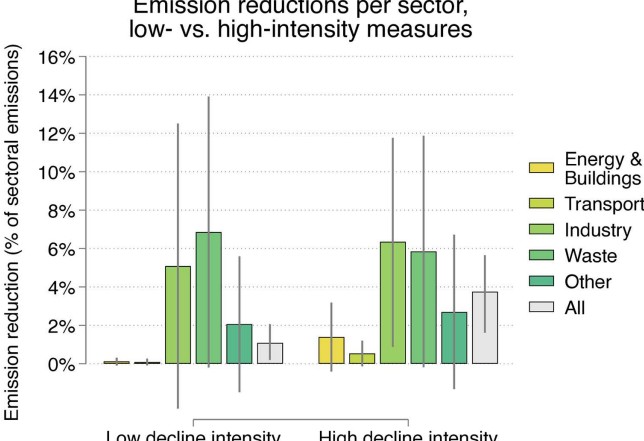

**Fig 8. Marginal means of CO$_2$ reduction potential by sector and policy type.** Bars show marginal means of the share of sectoral emissions avoided for 95 policies that included mitigation impact estimates in their Biennial Report, along with confidence intervals. A further three policies (not shown) were excluded due to implausibly high estimates that exceeded total annual sectoral emissions. Means are based on mixed-effects models, with random effects at the country level and robust standard errors. To calculate the normalised emission reductions, we used the highest mitigation impact estimate provided for any year between 2020 and 2040 in each country's Biennial Report and divided this by that sector's total annual emissions in 2019. The 'energy' category includes policies from the 'buildings' sector. Mitigation impact estimates for policies in the building sector were normalised in accord with the annual emissions of the energy sector because the former's emissions are not reported separately by countries. The 'other' category includes sectors with too few estimates to derive marginal means (agriculture, LULUCF, cross-cutting). 'All' represents the marginal mean obtained across all sectors. Decline intensity is categorised as 'low' (≤0.5) or 'high' (>0.5) based on intensity scores, which reflect the unweighted average of seven indicators (Table 4).

coverage may be more decisive in shaping a policy's potential mitigation impact than the choice between direct or indirect approaches. This indicates that both decline approaches – embodying contrasting functions of destruction and substitution – can meaningfully contribute to decarbonisation if well designed.

### 4.4. Limitations

We emphasise the tentative nature of our findings, as the statistical analysis draws on a limited number of observations – particularly in the LULUCF, transport and waste sectors. This limitation underscores the need for future research to

validate the observed relationships between decarbonisation effectiveness and decline intensity or the decline approach with a larger sample of policies and mitigation impact estimates.

A further methodological issue concerns the reliability of the mitigation impact estimates themselves, derived exclusively from figures provided by governments in their Biennial Report. Because the full range of assumptions and modelling methods underlying these estimates are not published, uncertainty remains regarding the comparability and accuracy of these forecasts of mitigation outcomes. Extending the analysis with third-party modelling based on standardised and more transparent approaches would offer an important means to validate or challenge these findings.

Additionally, our understanding of the decarbonisation impacts of different kinds of decline policies is hampered by the varying timescales over which GHG emission reductions occur. For example, policies that induce the reduction of coal-fired power typically achieve large and rapid short-term reductions. In contrast, policies aimed at reducing ICE vehicle sales typically result in slower emissions reductions due to the many years required to turn over existing on-road vehicle stock. To address this, our analysis used the highest estimate of annual mitigation impact provided in Biennial Reports regardless of the year in which this is expected to occur. Yet future research is needed to more fully account for the temporal dynamics of decline processes when comparing the relative decarbonisation effectiveness of different policy types.

## 5. Synthesis: Towards a typology of decline policies

The above findings revealed tremendous variation in the characteristics of decline policies. This provides an ideal opportunity to structure the policies in our dataset in heuristic fashion to arrive at a basic typology, shown in Table 7. Based on the core variables shown to be empirically significant in the preceding analyses, we suggest that decline policies can be broadly categorised according to their decline approach (direct or indirect) and decline intensity (high or low based on our indicators). *Decline approach* reflects whether a policy directly targets emission-intensive elements with a destructive function or whether it indirectly contributes to decline through a substitutive function by replacing incumbent components with cleaner alternatives. *Decline intensity* captures the extent to which a policy incorporates design features (see Table 4) expected to amplify the magnitude and speed of mitigation impact.

We underscore the qualitative nature of this typology. Unlike the above regression analysis, we do not set numerical thresholds here to distinguish 'high' from 'low' intensity. Furthermore, this classification is not derived from the dependent variable (mitigation impact), but rather from the two dependent variables (decline approach and decline intensity). As the preceding analysis demonstrates, these last two dimensions are not only highly valuable for characterising the decline policies in our dataset, but they also hold critical relevance for decarbonisation impacts. This said, since the primary goal of this exercise lies in capturing the diversity of decline policies designed by governments, we do not evaluate the emissions

**Table 7. A typology of decline policies and expected effect.**

| | | Decline intensity | |
|---|---|---|---|
| | | **High** | **Low** |
| **Decline approach** | Direct (decline through destruction) | **Type 1:** Abolishes the old, signalling where not to go, leveraging strictness, explicit targets, coverage and speed. Example policies (ID): coal phase-out or emission regulations (66, 68, 127, 227); fossil-heating installation bans (93, 170); emissions trading schemes (130, 169); logging bans (28). | **Type 3:** Reduces the old, but with limited strictness, reduction ambitions, scale, speed or coverage. Example policies (ID): carbon taxes (38, 220); fertiliser reduction support (121, 210); food waste reduction programmes (126). |
| | Indirect (decline through substitution) | **Type 2:** Substitutes old with new, signalling where to go, leveraging strictness, explicit targets, coverage and speed. Example policies (ID): ZEV sale targets (75, 115); renewable energy penetration targets (3, 24, 37, 250). | **Type 4:** Substitutes old with new, but with limited strictness, reduction ambitions, scale, speed or coverage. Example policies (ID): biomethane grid injection programmes (183); ICE vehicle replacement programmes (84, 114, 231); fuel switching subsidies (35, 43, 113). |

ID numbers refer to specific policies listed in the codebook (see S1 Data).

reduction effectiveness of each archetype. We leave this task to future research, ideally supported by a larger sample of policies and emissions reduction estimates.

In this section's remainder, we elaborate on the qualitative differences across the four decline policy types and critically consider their respective shortcomings and implementation challenges. This discussion is informed by earlier research [95] highlighting the limited transformative potential of substitution-focused approaches and the importance of embedding decline policies within broader strategies aimed at inducing wider system change.

*Type 1* decline policies seek to directly reduce the production or use of established technologies, materials and processes, typically through stringent and explicit guidance about 'where not to go'. Policies in this category simultaneously feature a higher decline intensity, reflecting the presence of regulatory instruments, ambitious numerical targets for the magnitude and speed of reduction, and coverage of broad areas across that country, market and technological lifecycle. Our analysis reveals that type 1 policies are frequently used to target technologies such as coal-fired power, fossil-fuel heating and ICE vehicles, substances such as ozone-depleting chemicals, and practices such as deforestation. In terms of inherent weaknesses, the conflictual nature of type 1 decline policies can trigger resistance by affected industries and communities [17,46]. Meanwhile, their implementation can be delayed or abandoned when alternatives are not sufficiently available for economic, technical or other reasons [96–98].

*Type 2* indirect decline policies primarily pursue a substitutive rather than destructive function, while similarly exhibiting a high level of decline intensity. Although these policies focus on directing actors toward appropriate alternatives, they contribute to decline by replacing or displacing the incumbent arrangement, creating space in a system for new creations to diffuse. Type 2 decline policies are best exemplified by mandates that set targets for ZEV sales as opposed to directly banning ICE vehicle sales or use – a hallmark of type 1 policies. As discussed, a strength of indirect policies is that they can prove more politically feasible than direct decline policies, which single out and seek to destroy 'bads' [3]. Conversely, a challenge associated with type 2 indirect decline approaches is that they contravene the prevailing, yet often misplaced, emphasis in policy and practice on avoiding 'picking winners' and maintaining technology neutrality [99]. If specific innovations are prioritised during substitution programmes, this concern may be strategically weaponised by stakeholders to resist type 2 policies on the grounds that they prematurely close off certain options while promoting others. Consider for instance the political resistance from German automakers to the ZEV-forcing effect of European-level regulation, triggering calls to extend the future of ICE vehicles through carbon-neutral e-fuels [11]. Another innate challenge for type 2 policies (and indeed type 4) is the risk of steering a market toward narrowly defined alternatives with limited transformative potential [100]. For instance, ZEV mandates in the light-duty vehicle segment are sometimes criticised as substituting ICE vehicles with battery-powered alternatives to the detriment of other more transformative options, such as shifting towards active and collective transport modalities [101,102]. Similarly, another potential pitfall is the risk of diffusing alternatives that bring decarbonisation or sustainability benefits in the short to medium term but create unanticipated consequences or lock-ins in the long-term [95,103]. This has indeed occurred as a result of early biofuel mandates in Europe, which drove deforestation in producing countries such as Indonesia and Malaysia, undermining the intended climate benefits [104].

*Type 3* and *type 4* policies – encompassing both direct and indirect approaches – are characterised by weaker decline intensity compared to the first two types. These policies often lack critical design features such as robust enforcement mechanisms, ambitious reduction objectives, explicit targets for the speed and magnitude of reduction, and broad sectoral coverage. A key concern with low-intensity type 3 and 4 policies is, therefore, their potential to delay fundamental change by perpetuating a partly decarbonised or scaled-down version of the existing carbon-intensive system [105,106].

In the case of type 3 decline policies, which encompass a destructive function but with weak intensity, many instances only aim to partially reduce incumbent arrangements. Mirroring type 1 policies, weaker destruction ambitions often reflect the absence of viable alternatives, as exemplified by industrial agriculture's continued reliance on chemical fertilisers. Notwithstanding, a significant risk associated with type 3 policies is that their outcomes may be limited to per-unit efficiency gains or marginal emissions reductions – such as a decrease in organic waste volumes per capita – rather than achieving

fundamental change or more transformative solutions [95]. In the case of policies aimed at technologies, if only affecting new installations, which leaves the existing high-emitting stock intact, the decline effect and mitigation outcomes can be compromised.

Similarly, *type 4* policies, characterised by both indirect decline and weak intensity, may result in open-ended reduction or even *stabilisation* of existing systems. Indeed, our dataset contains several policies mandating the blending of an increasing portion of renewable sources with conventional fuels in road and air transport. At first glance, such policies appear to facilitate the substitution of petroleum-based fuels with cleaner alternatives by virtue of their minimum blend targets. However, closer inspection reveals that renewable fuels are often layered on top of existing systems, merely diluting carbon-intensive fuels rather than catalysing a transformative shift in transport systems away from fossil fuels. In these cases, decline effects can be particularly compromised when substitution targets lack ambition, which consequently limits the extent of displacement that occurs.

## 6. Discussion and conclusion

This pilot study provides a first-of-a-kind empirical assessment of the decline policies employed by ten high-emitting countries under the Paris Agreement, contributing to the rapidly growing scholarly interest in the role of deliberate decline in driving transitions [3,4]. A key contribution lies in the development and examination of a dedicated database of decline policies, derived from descriptions of mitigation actions reported to the UNFCCC. Through this approach, we identified key sectors and targets of decline activity in diverse areas. Beyond this empirical mapping, our study advances a new conceptual and analytical toolkit for understanding and assessing decline policies. First, we conceptualised two broad approaches to pursuing decline: direct decline policies, which primarily focus on reducing incumbent arrangements and signalling 'where not to go', and indirect policies, which contribute to reduction by substituting old with new and signalling 'where to go'. Second, we inductively developed a 'decline intensity' scoring scheme to measure the presence of key design features expected to influence a policy's ability to achieve large-scale decline and mitigation outcomes. Third, we combined these two perspectives to propose a typology of four archetypical decline policies, categorising them based on their directness and decline intensity.

Our findings indicate that decline policies are widely used as a decarbonisation strategy under the Paris Agreement, evidenced by their prolific adoption across all countries and sectors in our sample, including beyond the energy realm. Moreover, both direct and indirect decline approaches are widely used, though marked differences occur in the relative prominence of each across countries and sectors.

Beyond this consideration of policy quantity and distribution, our conceptualisation of 'decline intensity' enhances understanding of the qualitative policy design features that influence the decarbonisation effectiveness of decline strategies. Critically, our analysis provides preliminary evidence that policy design – encompassing the decline approach (direct or indirect) as well as the various features that determine decline intensity – may play an important role in determining expected decarbonisation outcomes. Particularly, findings revealed that high-intensity policies are a robust predictor of larger mitigation impact estimates. This suggests that if policies are characterised by features such as broad geographical and sectoral coverage, explicit targets to stipulate the pace and magnitude of reduction, and ambitious timescales, decline interventions tend to translate into deeper decarbonisation compared to policies with less ambition. Furthermore, this observation holds true for both direct and indirect approaches, signalling that all types of decline policies can advance decarbonisation goals if designed appropriately. These findings help to conceptually and empirically clarify the intertwined nature of destruction and innovation in decline processes, along with how policies and specific design features can actively spur these [12,82].

The methodology developed for this paper offers a useful approach for tracking progress towards key mitigation outcomes across UNFCCC countries. Such an approach is increasingly critical amidst the growing momentum towards monitoring progress in phasing out carbon-intensive technologies. This trend is evident in the International Energy Agency's

efforts to track headway to achieving fossil-fuel reduction and clean technology upscaling commitments made at COP28 [7] and in the call for a global fossil fuel phase-out by climate scientists [107]. Our comprehensive database of decline policies also complements existing resources, such as the University of Sussex's 'Supply-side Policy Tracker', which offers researchers and practitioners an integrated overview of various decline instruments, including bans and moratoria on fossil-fuel extraction.

In closing, two chief limitations of this study represent promising avenues for future research. First, the low number of countries and limited availability of estimates of mitigation impact in our sample point to the need for broader analyses covering more countries, sectors and policies. This study relied exclusively on policy descriptions in Biennial Reports submitted to the UNFCCC as its chief data source. Since these reports are unlikely to comprehensively list the entire suite of a country's relevant decline policies – especially at the sub-national level – future work could extend our analytical procedure by drawing on additional data sources. These might include databases maintained by organisations such as IEA, IRENA and the NewClimate Institute as well as national or sub-national climate action plans. Future work could also move beyond Annex I countries to build a better picture of how decline policies feature in the decarbonisation efforts of geographies located in the Global South. Second, an important opportunity remains to investigate how political economy conditions shape a country's adoption and implementation of decline policies as a decarbonisation strategy. While our analysis revealed that some countries such as Germany feature a relatively high proportion of direct decline policies in their Biennial Reports, it did not explore the contextual reasons behind the cross-national variation in the shares of direct and indirect approaches. Similarly, questions remain about the conditions that affect the ability of countries to design and implement high intensity decline policies. Clarifying the political economy factors that shape the variations observed in policy characteristics would be an important way to enhance our understanding of the conditions needed to achieve politically accelerated decline.

## Supporting information

**S1 Data. Dataset, coding results and data for regression.**
(XLSX)

**S2 Code. Strata code used to run the regression.**
(DO)

**S3 File. Additional tables and figures.**
(PDF)

## Acknowledgments

The authors would like to thank Florentine Koppenborg in addition to Johanna Kuenzler, Mitya Pearson and participants at the General Conference of the European Consortium for Political Research, held in Dublin 2024, for valuable comments regarding an earlier draft. We also thank the reviewer for their constructive comments and enthusiastic engagement with our empirical material.

## Author contributions

**Conceptualization:** Gregory Trencher, Mert Duygan, Adrian Rinscheid, Daniel Rosenbloom, Peter Newell.

**Data curation:** Gregory Trencher, Adrian Rinscheid.

**Formal analysis:** Gregory Trencher, Mert Duygan, Adrian Rinscheid.

**Investigation:** Gregory Trencher.

**Methodology:** Gregory Trencher.

**Resources:** Gregory Trencher.

**Supervision:** Mert Duygan, Daniel Rosenbloom, Peter Newell.

**Validation:** Gregory Trencher, Adrian Rinscheid.

**Visualization:** Gregory Trencher, Adrian Rinscheid.

**Writing – original draft:** Gregory Trencher, Adrian Rinscheid, Daniel Rosenbloom.

**Writing – review & editing:** Gregory Trencher, Mert Duygan, Adrian Rinscheid, Daniel Rosenbloom, Peter Newell.

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
