## [Decision Letter · Decision Letter 0]

24 Apr 2025

Dear Dr. Rosenbloom,

Thank you for submitting your manuscript to PLOS ONE. After careful consideration, we feel that it has merit but does not fully meet PLOS ONE’s publication criteria as it currently stands. Therefore, we invite you to submit a revised version of the manuscript that addresses the points raised during the review process.

We look forward to receiving your revised manuscript.

Kind regards,

Syed Ahsan Ali Shah

Academic Editor

PLOS ONE

Journal Requirements:

https://journals.plos.org/plosone/s/file?id=ba62/PLOSOne_formatting_sample_title_authors_affiliations.pdf.m

2. Please update your submission to use the PLOS LaTeX template. The template and more information on our requirements for LaTeX submissions can be found at http://journals.plos.org/plosone/s/latex .

Reviewers' comments:

Reviewer's Responses to Questions

**Comments to the Author**

1. Is the manuscript technically sound, and do the data support the conclusions?

Reviewer #1: Yes

2. Has the statistical analysis been performed appropriately and rigorously?

Reviewer #1: No

3. Have the authors made all data underlying the findings in their manuscript fully available?

Reviewer #1: Yes

4. Is the manuscript presented in an intelligible fashion and written in standard English?

Reviewer #1: Yes

Reviewer #1: Thank you for the opportunity to review the manuscript “Decline by design: Assessing the role of policies that downscale carbon-intensive artefacts and activities under the Paris Agreement.”

The authors present a timely exploration of policies designed to reduce the production and use of carbon-intensive technologies across ten countries. They classify decline policies by type and analyze their expected impacts on emissions reductions. The manuscript is generally well written and tackles a highly relevant topic with clear policy significance.

However, the manuscript has several important shortcomings that must be addressed before it can be considered for publication.

1. Unclear knowledge gap.

The authors claim to help bridge three knowledge gaps in the literature, but they simply list characteristics of previous studies that differ from theirs instead of a genuine knowledge gap.

First, the authors mention that the literature on decline has mainly focused on historical cases. While I agree to some extent—some references (like [8], but also Plötz et al. 2018, and many others) address ongoing decline cases—it is unclear what can be learned that is new by looking at more recent cases.

Second, what “advancement of knowledge” is hindered by the “lack of systematic and comprehensive analyses of large volumes of policies” (page 3)? What is the knowledge gap here?

Third, I am not sure that because someone has not yet used a specific data source it can be considered a knowledge gap. The authors seem to simply repeat the same gap as they state in the previous paragraphs.

Put simply, what is the gap in our knowledge that your paper is bridging? What can be learned from using the new data and comparing policies across countries and sectors?

I believe there is value in the research the authors present, but the knowledge gap has to be motivated much more strongly.

Plötz, P., Axsen, J., Funke, S.A. and Gnann, T., 2019. Designing car bans for sustainable transportation. Nature Sustainability, 2(7), pp.534-536.

2. Conceptual Clarity.

In my opinion, the paper currently lacks a consistent and well-defined conceptual framework. Several key terms and categories remain ambiguous, hindering the manuscript's clarity and coherence. In particular:

2.1 Definition of "decline."

The authors refer to the "downscaling" of carbon-intensive technologies, materials, and processes, but they do not define what exactly is meant by "downscaling." This term requires a clear definition. To the best of my ability, the cited literature does not provide a definition either (e.g., refs 2, 3, 4, 7). What exactly is downscaling? Downscaling can be confused with moving from large-scale, centralized production to small-scale, distributed production. I would recommend that the authors stick to an established definition of decline.

2.2 What declines?

The authors claim to expand the conception of what can be targeted by decline policies, but do so in a very confusing way, suggesting that technologies, substances, practices, and institutions are what can decline (page 5). Lacking a clear conceptual framework, it is difficult to understand how these terms differ from each other and how they are interrelated. From a conceptual perspective, the reader is left uncertain about the scope and boundaries of the analysis. This unclarity has subsequent effects when the authors refer to “incumbent elements” and “incumbent arrangements” that are equally unclear.

I strongly suggest that the authors consider adopting one of the frameworks commonly used in the decline literature, like socio-technical systems, production and consumption systems, technological innovation systems, or others. Besides helping clarify what declines and allowing the authors to define and locate the different aspects of the system that decline policies can target, the use of a system-level framework would provide authors with an unambiguous vocabulary, for example, to talk about “incumbent regimes.” Last but not least, a clear conceptual framework would help the authors to discuss more intelligibly the processes through which the policies they study achieve their effects. In repeated occasions throughout the text, the authors refer to the different “levers” that decline policies can use. However, without a conceptual framework that explains how policies affect the different components of the systems they try to influence, it is unclear what these levers are, how different policy instruments can act on them, or why some design features should be expected to be more effective than others.

2.3 Indirect decline.

The conceptualization of indirect decline appears to be limited to substitution (e.g., promoting clean alternatives), but there could be other paths that lead to the reduction of the production and use of a technology without a direct substitution. For example, policies promoting working from home can lead to a reduction in the use of fuel because people travel less without supporting a cleaner alternative. Similarly, policies aimed at energy conservation can lead to a reduction in the use of fuel. Policies for building renovations could reduce the use of fossil fuels by improving insulation, even if not necessarily supporting a cleaner heating fuel alternative. Some of these policies appear in your dataset (e.g., policy 92) but do not seem to fit in your current description of indirect decline (on page 7). Some policies might even fall somewhere in between, for instance, how do you deal with efficiency or emission standards? If an efficiency or emission standard is set so high that the incumbent technology cannot meet it, does it classify as a direct or indirect decline policy?

3. Methodological concerns.

There are several aspects of the methodology that require clarification and more rigorous justification:

3.1 Potential selection bias?

On page 11, the authors state that policies using “decline-related terminology” were excluded if they were deemed to focus on diffusion or marginal decline. This approach appears arbitrary and could introduce selection bias. It might exclude policies that are introduced as decline measures but fail to achieve meaningful impacts. The authors should explain more clearly how policies were included or excluded from the sample. As a robustness check, it would be interesting to see a sample where any policies that explicitly mention a decline-related term in their description are added to the sample. With a list of the terms used to identify the policies.

3.2 Variable definitions.

The manuscript should clearly list: all sectors included in the study, the criteria used to classify policies as direct or indirect, all policy instruments considered, and their definitions. Each of these elements should be explicitly defined and justified. I could only find examples in the supplementary information rather than complete lists and definitions.

3.3 Data transparency.

The underlying data should be presented more transparently, ideally in tabular or supplementary format. This would improve reproducibility and allow readers to assess the robustness of the analysis more accurately. In particular, the authors must show correlation checks across the variables included in the regression analysis.

3.4 Other methodological concerns.

• Comparing emission reductions in absolute terms across countries and sectors of widely ranging sizes can lead to misleading results. Should not relative changes in emissions should be used (e.g., (emissions in sector A of country B at year X – emissions in sector A of country B at year X+t with policy) / emissions in sector A of country B at year X)?

• Why are differences between countries never shown in figures or tables?

• Figure 6 provides very little information. Show the distribution of values with boxplots or a similar graph.

• Figure 7 seems to mix direct and indirect policies.

• Figure 8 must show the uncertainty interval around the estimates.

4. Policy types

While the policy types sound relatively convincing, an analysis of the expected emission reductions suggests that there is no statistically significant difference between them (except between types 1 and 4). This makes me think that the authors may be overstating the relevance of their distinction or that further thinking or analysis must go into finding categories to differentiate decline policies that align better with their expected effectiveness. Here, I used absolute CO2 emissions reductions, but I believe the authors should instead use relative emissions reductions by sector and country. (I copy the code from Python in case they want to reproduce the test, a figure and a table in the attached PDF)

**Do you want your identity to be public for this peer review?** For information about this choice, including consent withdrawal, please see our Privacy Policy

Reviewer #1: No

---

## [Author Response · Author response to Decision Letter 1]

9 Jul 2025

As our responses included graphical information, we have uploaded our replies to reviewer comments as a dedicated file. Please see the included file.

---

## [Decision Letter · Decision Letter 1]

30 Sep 2025

Decline by Design: Assessing decline policies as a decarbonisation strategy under the Paris Agreement

PONE-D-24-56415R1

Dear Dr. Rosenbloom,

We’re pleased to inform you that your manuscript has been judged scientifically suitable for publication and will be formally accepted for publication once it meets all outstanding technical requirements.

Kind regards,

Tingzhen Ming, Ph.D.

Academic Editor

PLOS ONE

Additional Editor Comments (optional):

Reviewers' comments:

Reviewer's Responses to Questions

**Comments to the Author**

Reviewer #1: All comments have been addressed

2. Is the manuscript technically sound, and do the data support the conclusions?

Reviewer #1: Yes

3. Has the statistical analysis been performed appropriately and rigorously?

Reviewer #1: Yes

4. Have the authors made all data underlying the findings in their manuscript fully available?

Reviewer #1: Yes

5. Is the manuscript presented in an intelligible fashion and written in standard English?

Reviewer #1: Yes

Reviewer #1: Dear Authors,

Thank you very much for addressing my comments on your manuscript in such great detail. I agree with you and believe that the manuscript has improved significantly. In my view, the revised conceptual framework has particularly improved.

Overall, I am sufficiently satisfied with how you have addressed my concerns.

**Do you want your identity to be public for this peer review?** For information about this choice, including consent withdrawal, please see our Privacy Policy

Reviewer #1: No

---

## [Editor Report · Acceptance letter]

PONE-D-24-56415R1

PLOS ONE

Dear Dr. Rosenbloom,

I'm pleased to inform you that your manuscript has been deemed suitable for publication in PLOS ONE. Congratulations! Your manuscript is now being handed over to our production team.

Kind regards,

on behalf of

Dr. Tingzhen Ming

Academic Editor

PLOS ONE